



# Going Beyond BEM with BEM: an Insight into Dynamic Inflow Effects on Floating Wind Turbines

Francesco Papi[1], Jason Jonkman[2], Amy Robertson[2], Alessandro Bianchini[1]

[1]Department of Industrial Engineering, Università degli Studi di Firenze, Florence, 50139, Italy
[2]National Renewable Energy Laboratory, Golden, CO, USA

*Correspondence to*: Francesco Papi (fr.papi@unifi.it) or Alessandro Bianchini (alessandro.bianchini@unifi.it)

**Abstract.** Blade Element Momentum (BEM) theory is the backbone of many industry-standard wind turbine aerodynamic models. To be applied to a broader set of engineering problems, BEM models have been extended since their inception and now include several empirical corrections. These models have benefitted from decades of development and refinement and have been extensively used and validated, proving their adequacy in predicting aerodynamic forces of horizontal axis wind
turbine rotors in most scenarios. However, the analysis of Floating Offshore Wind Turbines (FOWTs) introduces new sets of challenges, especially if new-generation large and flexible machines are considered. In fact, due to the combined action of wind and waves and their interaction with the turbine structure and control system, these machines are subject to unsteady motion, and thus unsteady inflow on the wind turbine's blades, which could put BEM models to the test. Consensus is not present yet on the accuracy limits of BEM in these conditions. This study contributes to the ongoing research on the topic by
systematically comparing four different aerodynamic models, ranging from BEM to Computational Fluid Dynamics (CFD), in an attempt to shed light on the unsteady aerodynamic phenomena that are at stake in FOWTs and whether BEM is able to model them appropriately. Simulations are performed on the UNAFLOW 1:75 scale rotor during imposed harmonic surge and pitch motion. Experimental results are available for these conditions and are used for baseline validation. The rotor is analysed both in rated operating conditions and in low wind speeds, where unsteady aerodynamic effects are expected to be more
pronounced. Results show how BEM, despite its simplicity, if augmented with a dynamic inflow model, is able to adequately model the aerodynamics of FOWTs in most conditions.

## 1 Introduction

Within the wind energy research community, many are questioning the validity and applicability, especially in the context of
FOWTs, of the aerodynamic theories used in medium-fidelity engineering tools commonly applied to FOWT design. Aerodynamic models based on Blade Element Momentum (BEM) Theory are still at the core of many design codes (Veers et al., 2022). With certain assumptions, BEMT allows for the effects of the wind turbine's wake on the inflow local to the rotor blades to be estimated. Thereafter, aerodynamic loads can be determined using blade element theory, i.e., assuming that each blade section behaves as a 2D airfoil. The added value of BEMT is that it allows a fundamental understanding of the effects



of varying geometrical and aerodynamic parameters on a wind turbine (Leishman, 2016). It also works very well in practice, which is undoubtedly important in engineering. The limitations of this aerodynamic theory are however apparent. As explained in detail in the following, various engineering models have been developed and implemented into BEM-based aerodynamic models to extend their range of validity. For instance, empirical corrections have enabled the extension of these models to compute aerodynamic loads in the vortex ring state (VRS) or turbulent wake state (TWS), as detailed in (Sørensen et al., 1998).

Empirical corrections to model tip and root losses, non-uniform inflow, unsteady inflow and skewed flow are also introduced into most design-level BEM-based codes. Critical comparisons of medium-fidelity aerodynamic theories on onshore rotors have been performed in the past, examples of which are the studies by Perez-Becker et al. (Perez-Becker et al., 2020) and Boorsma et al. (Boorsma et al., 2020). While BEM-based models have been found to perform adequately, some deficiencies in the prediction of cyclic loads are highlighted.

Floating wind turbines introduce additional challenges from an aerodynamic standpoint as the rotor is subject to unsteady motion. Some authors have highlighted how rotor-wake interaction is possible in a FOWT due to the rotor rocking in and out of its own wake (Veers et al., 2022). This phenomenon was observed in numerical simulations by (Sebastian and Lackner, 2013) and (Ramos-García et al., 2022), when simulating high frequency and high amplitude platform oscillations in the wind heading direction - i.e. surge or pitch oscillations. This is seen as a source of concern because, if rotor-wake interaction occurs,

the streamtubes, upon which the momentum balance is applied, are effectively chocked and momentum theory is invalid. Moreover, some have come to the conclusion that, in these conditions, the rotor may enter wake states where momentum theory is invalid, such as the TWS or VRS, for which empirical corrections have previously been developed. The underlying assumption when applying momentum theory to a rotor is that it is in an inertial reference frame. Momentum balance is applied in the rotor reference frame, and because the rotor is static or moving with constant speed (this could be the case for a helicopter

rotor), rotor movement is treated the same way as wind speed is, as its only effect is to introduce an apparent wind speed on the blades themselves. When the rotor motion becomes unsteady, the reference frame is not inertial and a momentum balance cannot be performed in such a reference frame. Based on this consideration, Ferreira et al. (Ferreira et al., 2022) proposed to apply the momentum balance for a FOWT in the static reference frame and developed a correction to account for the dynamic inflow. On the other hand, many other BEM-based codes simply ignore this hypothesis and perform the momentum balance

in the rotor reference frame, regardless of whether it is moving or not. Despite this being theoretically inaccurate, some have noticed better agreement with experiments and higher fidelity methods when the rotor apparent velocity due to its motion is included in the momentum balance, rather than it being accounted for only in the blade-element part of the BEM balance (Boorsma and Caboni, 2020).

It can be argued, however, that applying the momentum balance in the rotor, non-inertial frame is in practice no different from

considering unsteady inflow on the rotor. This condition is routinely treated in BEM models, and engineering corrections for dynamic inflow exist to extend the validity of BEM to these conditions. Moreover, although momentum theory is invalid when the rotor enters TWS of VRS, empiric corrections such as Glauert's correction for TWS (Buhl, 2005) are implemented into engineering codes. Therefore, if the rotor enters such states as a consequence of unsteady motion and the way the momentum





balance is applied, it is the validity of these engineering corrections that should be challenged, rather than momentum theory

itself.

The performance of momentum theory and its empirical correction models as implemented in a state-of-the-art wind turbine simulation tool is discussed in this paper. BEM results computed on the UNAFLOW (Bernini et al., 2018) 1:75 scale rotor are compared to Lifting-line Free Vortex Wake (LLFVW) and Actuator Line Model (ALM) results. Experimental results (Fontanella et al., 2021; Mancini et al., 2020) are used for validation when available. The numerical models, with their

respective advantages and shortcomings, and test case are introduced in Section 2. Section 3 presents results of tests with surge oscillations during rotor operation at rated wind speed, followed by pitch oscillations in cut-in wind conditions, where, we argue, rotor-wake interaction is most likely to occur. Section 4 draws some conclusions of the study and proposes a new perspective on the real capability of the BEMT theory to deal with the complex phenomena taking place in FOWTs.

## 2 Methods

In this section, the main details of the aerodynamic theories that are compared in this work, as they are implemented in the codes used in this comparison, are explained. The most important details of the experimental apparatus and test-case that is used in the code-to-code comparison endeavor and the examined test cases are also described.

### 2.1 Numerical Models

Four different aerodynamic models are compared in this work, namely BEM, Dynamic-BEM (DBEM), LLFVW and ALM.

All the tested models are implemented within the common framework of OpenFAST v3.1.0 (OF). In particular, the standalone version of OF's aerodynamic module AeroDyn is used. An additional aerodynamic model, more specifically the ALM model, is implemented directly in AeroDyn's glue-code OpenFAST instead. Despite this difference, blade aerodynamics are handled by the same module, AeroDyn, in all cases. In more detail, the main characteristics of the numerical models are described in the following.

**Blade Element Model** To isolate the differences between the wake modelling theories, all the tested aerodynamic theories, regardless of the strategy used to model the wake, use the same blade model. Each blade is divided into twenty segments, small enough so to consider the blade geometry constant within each of them. Tabulated lift and drag coefficients computed for various Reynolds numbers and corrected for 3D effects are assigned to each section. Once the relative velocity – dependent on the wake model and inflow conditions – is known, the aerodynamic lift ($F_l$) and drag ($F_d$) forces can be computed for each

section as $F_{l,d} = 1/2\,\rho c C_{l,d} U_{rel}^2$, where $C_{l,d}$ is the lift or drag coefficient, $\rho$ is the air density, $c$ is the blade chord and $U_{rel}$ is the relative wind speed to each blade section. Aerodynamic coefficients for the UNAFLOW rotor are derived from wind-tunnel measurements, as described in (Fontanella et al., 2021). In addition, some of the wake models are tested using the tabulated Cl and Cd (*static polars* in the following), while others are also tested with Gonzalez's variant of the Beddoes-Leishman dynamic stall and unsteady airfoil aerodynamic model (Damiani and Hayman, 2019) (*dynamic polars*). Unsteady



airfoil aerodynamic effects, including those that occur in the attached-flow regime, grow increasingly larger as the airfoil reduced frequency (for an isolated oscillating airfoil it is defined as $f_{rc} = \pi f c / U_\infty$, where $f$ is the oscillation frequency and $U_\infty$ is the incoming wind speed) increases (Leishman, 2016). In this work, inflow angles are mostly kept below stall, and thus attached flow unsteady aerodynamic effects have the largest impact on results. These are mainly caused by two effects: added mass and shed vorticity, with the latter being by far the most relevant at the reduced frequencies analysed in this work. The

widespread consensus is that the latter are intrinsically included in higher-order aerodynamic theories such as LLFVW and ALM and do not require dynamic polars to model this effect.

**BEM** The first aerodynamic theory that is compared is Blade-Element Momentum Theory, referred to in the following simply as BEM. A full theoretical overview and background of this theory can be found in (Burton, 2001; Hansen, 2008). In this work the authors assume the reader is familiar with this aerodynamic theory and only the main details of the specific BEM

implementation used in this work will be introduced. More specifically, the guaranteed BEM solution algorithm developed by Ning (Ning, 2013) and implemented in AeroDyn v15 (Ning et al., 2015) is used herein. The three blades are divided into a series of segments and the momentum balance is applied for each segment of each blade separately. Notably, this deviates from the classical BEM approach of dividing the rotor into a series of annular streamtubes and applying the momentum balance on each streamtube. Prandtl's tip and root loss corrections are included directly in the momentum balance as well as Glauert's

high-induction empirical correction, as adapted by Buhl (Buhl, 2005). In addition, structural velocities are considered within the momentum balance directly. Thus, although equations in this form will not be explicitly found in the AeroDyn code, the axial momentum balance for each blade node $i$ can be written as:

$$U_{relX}^i = (1 - a^i)(U_{windX}^i + U_{strX}^i) \tag{1}$$

Therefore, in case of a complex motion, such as a platform pitching action, a separate momentum balance is applied for each

blade node with different relative velocities. It is important to point out that Eq. (1) is for discussion purposes only, and refers exclusively to axial velocity, neglecting the contribution of tangential induction. After the momentum balance is performed, the axial induction is corrected to account for the effect of skewed inflow, using the Pitt-Peters skewed inflow model (Ning et al., 2015). It is immediately apparent to the more experienced reader how this theory deviates significantly from the basic Blade Element Momentum theory, that is more rigorously formulated and valid for a static, infinite-blade actuator disk subject

to constant and uniform inflow, aligned with the rotor disk itself. Moreover, some of the empirical corrections that were developed in time to extend the range of applicability of BEM-based models, such as Glauert's high-induction correction, are technically valid over the entirety of an actuator disk, and there is no theoretical guarantee that they can be applied to a single blade element, as discussed by Buhl (Buhl, 2005). In addition to these inconsistencies between BEM theory and BEM-based aerodynamic models, in the context of FOWTs, the validity of Eq. (1) is debated. In fact, a momentum balance can only be

performed in an inertial reference frame in steady-state inflow and operating conditions. Therefore Eq. (1) is strictly valid only when $U_{str}^i = const$ and $U_{wind}^i = const$. In practice however, this is often ignored, and Eq. (1) is applied to non-inertial FOWTs as well, leading at times to erroneous considerations regarding the behaviour of these systems. This issue was recently





raised by Ferreira et al. (Ferreira et al., 2022). Despite this approach being theoretically incorrect, however, Boorsma and Caboni (Boorsma and Caboni, 2020) found that including platform velocity in the momentum balance, as if the balance was

performed in the non-inertial reference frame of the FOWT rotor, improved the agreement with higher-order models and experiments during simplified tests. Mancini et al. (Mancini et al., 2022) discussed the issue further, and found improved agreement with higher order aerodynamic theories also in forced pitch and yaw tests. According to the authors (Mancini et al., 2022), these models fully qualify as engineering models or "corrections", as they are based on empiric observations and not do not possess a solid theoretical basis. Indeed, these models, including those tested herein, are only loosely related to BEM

theory, and thus qualify as "BEM-based" models. In this work, results and discussion are relative to BEM and DBEM models in which the momentum balance is performed in the rotor reference frame.

Finally, while some of these considerations, such as performing a separate momentum balance for each blade, are valid only for the specific BEM implementation at hand, others, such as the application of Glauert's high-induction correction, are common practice within the industry and can be thus considered more general.

**DBEM** In addition to the BEM theory discussed in the previous paragraph, Dynamic-BEM introduces an additional empirical correction to account for dynamic inflow effects. The dynamic correction model used herein is Øye's two-equation low-pass filtering approach (Snel and Schepers, 1995). Dynamic inflow models have originally been developed to capture the rotor load over or under shoots that were observed in the case of a step-change in rotor speed or blade pitch. Since most of these models are empirical, there is no guarantee that they are able to reproduce the wake dynamics in the case of FOWT motions, when

blade inflow is continuously changing. For the more interested reader, a detailed discussion regarding dynamic inflow models for FOWTs is provided in (Ferreira et al., 2022).

**LLFVW** Lifting-Line Free Vortex Wake is a medium-fidelity wake model. The LLFVW model implemented within AeroDyn, cOnvecting LAgrangian Filaments (OLAF) (Shaler et al., 2020), is used in this work.

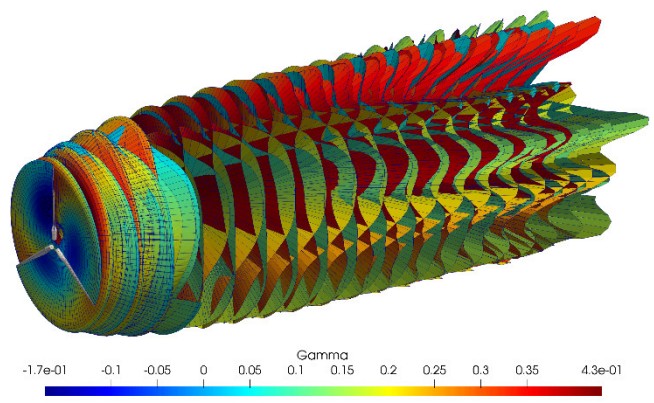

**Figure 1: Representation of the vortex elements in the wind turbine wake in the LLFVW model. Vortices are coloured by circulation values.**



The blades are modelled as a series of vortex rings. At each timestep the rings are shed into the wake and form a vortex lattice, as shown in Fig. 1. As explained in detail in (Van Garrel, 2003), the circulation of the vortex rings is found by equating the blade element forces to the Kutta–Joukowski theorem. The main model settings are as follows. Near wake is free, two
revolutions long, and modelled with full shed and trailing vorticity. Far wake is modelled by lumping the vorticity into the tip and root vortices only. It is six revolutions long and also free. Induced velocities are computed using Biot-Savart's law. The Vatistas core-vortex model is used, initial vortex core radius is set to be 5% of the local chord. Core radius is increasing with vortex age, depending on viscous diffusion $\delta$, set to 100 in this study after initial calibration:

$$r_c(\varsigma) = \sqrt{r_{c0}^2 + 4\alpha\delta\upsilon\varsigma} \qquad (2)$$

LLFVW models are widely acknowledged as an improvement over BEM theory. Although they come with an increased computational burden, they are able to intrinsically model effects such as skewed, dynamic non uniform inflow and root and tip losses without the need for additional empiric corrections.

**ALM** The concept of an Actuator Line Model for wind turbines was first proposed by Sorensen and Shen (Sorensen and Shen, 2002), and allows the wind turbine wake to be resolved using Navier-Stokes equations (i.e., Computation Fluid Dynamics),
with limited computational cost with respect to a full CFD solution. The basics of an ALM model can be described as follows. The wind turbine blades are divided into a series of blade sections, for which 2D characteristics (*Cl*, *Cd*) are determined, as is the case for other *"lifting-line"* based approaches such as those described in the previous paragraphs. For each blade section, the relative velocity $W$ is determined by combining the velocity that is sampled from the CFD domain and the structural velocity as $\vec{W} = \vec{U}_{CFD} + \vec{U}_{ST}$. This process is commonly referred to as velocity sampling. In this work, the ALM code
developed by coupling OpenFAST to the CFD solver CONVERGE (Richards et al., 2023), CALMA (Converge Actuator Line Model for Aeroelasticity) (Pagamonci et al., 2023) is used. In this code, velocity sampling is handled through the line-average velocity sampling algorithm proposed by Jost et al. (Jost et al., 2018). Once the blade forces are computed, they are inserted into the CFD domain as body forces through the force projection procedure. In this work, a piecewise function is used to smear the forces into the domain, as proposed by Xie (Xie, 2021). The kernel size is a trade-off between numerical stability and
accuracy; a kernel size equal to one fourth of the chord length is used in the inner 60% of the blade, while in the outer part of the blade, where chord size is smaller, kernel size is limited by the cell dimension to be four times the local cell size. The computational domain is shown in Fig. 2 and matches the dimensions of the Politecnico di Milano (PoliMi) wind tunnel as closely as possible. On the wind tunnel walls, a slip condition is imposed to avoid resolving the wall boundary layer. To account for the latter, the walls are moved inward by a distance equal to the boundary later height, as estimated during
preliminary experimental calibration (Bernini et al., 2018). The grid is cartesian with successive levels of refinement close to the rotor. The base grid size is 0.25 m, in proximity of the rotor the grid size is $0.25 * 2^{-4} = 0.015625\ m$. In the wake, cell size is $0.25 * 2^{-3} = 0.03125\ m$ and Automatic Mesh Refinement (AMR) is used to increase cell size where needed. This approach has shown to work well and was calibrated with respect to experiments, as shown in the results presented by Bergua





et al. (Bergua and et. al., 2023) and is able to resolve the tip vortex trajectory with accuracy similar to that of comparable
numerical techniques (Cioni et al., 2023).

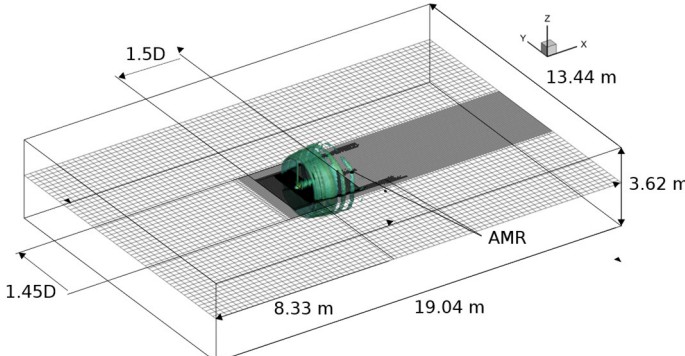

**Figure 2: ALM model setup.**

**2.2 Test case**

The UNAFLOW rotor is an approximately 2.4 m diameter rotor developed by PoliMi during the LifeS50+ project and later
used in the UNAFLOW test campaign (Bernini et al., 2018) shown in Fig. 3. The rotor is a 1:75 scale version of the DTU 10-
MW rotor (Bak et al., 2013). The blade was re-designed to match the aerodynamic characteristics of the full-scale rotor as
closely as possible, with focus being placed especially on matching thrust coefficients, as explained in detail in (Bayati et al.,
2017b, 2016).

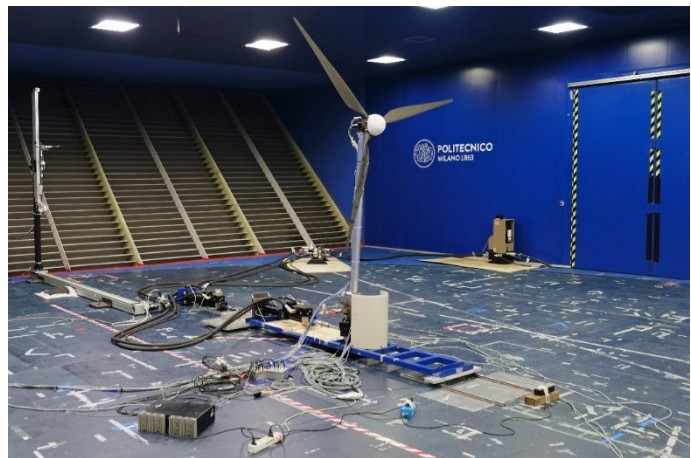

195       **Figure 3: UNAFLOW Rotor during tests in the Politecnico di Milano wind tunnel. [Photo Credit: Politecnico di Milano].**

The rotor diameter is geometrically scaled, but a low-Reynolds airfoil was used along the entire blade span and, crucially for
this study, the local chord to rotor diameter ratio was increased along most of the blade span. The wind speed imposed in the
wind tunnel tests is 1/3 of the full-scale wind speed. During the UNAFLOW test campaign, various oscillatory surge and pitch
tests of the rotor were conducted. The amplitude of the test article's oscillations are scaled geometrically, while the frequency
at model scaled is obtained by imposing a constant reduced frequency, defined by Bayati et al. (Bayati et al., 2017a) as $f_r =$



$\frac{fD}{U}$. Results are summarized in the work of Mancini et al. and Fontanella et al. (Fontanella et al., 2021; Mancini et al., 2020). The outcomes of the experiments were later made available to the participants to IEA wind task 30 (OC6 phase III), and used for code-to-code comparisons and validation (Bergua and et. al., 2023). In this work, the data made available to the IEA Wind Collaborative Task 30 ("OC6") participants is used as a basis to check the soundness of the numerical results with respect to

the experimental baseline. We then build on these test cases and perform forced surge and pitch simulations at frequencies and amplitudes that go beyond what is possible with the PoliMi experimental apparatus. Numerical tests in low wind speeds are also performed, to verify model predictions near rotor cut-in wind speed. The load cases (LCs) that are included in this study are summarized in tables 1 and 2. For each LC, the combinations of frequency, amplitude, wind speed and rotor speed are shown. In addition, the reduced frequency and full-scale amplitudes and frequencies of the tests are highlighted.


**Table 1: Surge Load Cases (LCs) discussed in this study. In bold, LCs for which experimental measurements are available. In italic, LCs used in OC6 Phase III (Bergua and et. al., 2023).**

| LC | *2.1* | *2.5* | *2.7* | 2.20 | 2.21 | 2.28 | 2.29 | 2.31 | *2.12* | *2.16* | *2.17* |
|---|---|---|---|---|---|---|---|---|---|---|---|
| f [Hz] | *0.125* | *1* | *2* | 3 | 4 | 8 | 12 | 16 | *2* | *2* | *2* |
| A [m] | *0.125* | *0.035* | *0.008* | 0.008 | 0.008 | 0.008 | 0.008 | 0.008 | *0.08* | *0.08* | *0.08* |
| $U_\infty$ [m/s] | *4.19* | *4.19* | *4.19* | 4.19 | 4.19 | 4.19 | 4.19 | 4.19 | *4.19* | *4.19* | *4.19* |
| $f_r$ [Hz] | *0.071* | *0.568* | *1.137* | 1.705 | 2.273 | 4.547 | 6.820 | 7.957 | 1.137 | 1.137 | 1.137 |
| rpm | *240* | *240* | *240* | 240 | 240 | 240 | 240 | 240 | *240* | *240* | *240* |
| $\Delta$rpm/$\Delta\theta$ | - | - | - | - | - | - | - | - | - | *±36 rpm* | *±1.5°* |
| $f_{fs}$ [Hz] | *0.005* | *0.04* | *0.08* | 0.12 | 0.16 | 0.32 | 0.48 | 0.64 | *0.08* | *0.08* | *0.08* |
| $A_{fs}$ [m] | *9.375* | *2.625* | *0.6* | 0.6 | 0.6 | 0.6 | 0.6 | 0.6 | *6* | *6* | *6* |
| $V_{max}$ [m/s] | *0.098* | *0.220* | *0.101* | 0.151 | 0.201 | 0.402 | 0.603 | 0.804 | *1.01* | *1.01* | *1.01* |

**Table 2: Pitch Load Cases (LCs) discussed in this study.**

| LC | 3.26 | 3.27 |
|---|---|---|
| f [Hz] | 2.5 | 2.5 |
| A [m] | 2 | 1 |
| $U_\infty$ [m/s] | 1.6667 | 1.6667 |
| $f_r$ [Hz] | 0.187 | 0.187 |
| rpm | 150 | 150 |
| $\Delta$rpm/$\Delta\theta$ | - | - |
| $f_{fs}$ [Hz] | 0.1 | 0.1 |
| $A_{fs}$ [m] | 2 | 1 |
| $V_{max}$ [m/s] | 1.452 | 0.726 |





## 3 Results

The results section is organized as follows. Aerodynamic force prediction capability of the aerodynamic theories in nominal operating conditions is discussed first. These conditions correspond to the test cases that were evaluated during the OC6 Phase III project (Bergua and et. al., 2023), but additional insight is provided herein. Oscillatory pitch tests at cut-in wind speed, where rotor loading is high, are discussed next. Finally, FOWT wake states are discussed.

### 3.1 Rated wind speed

Rotor thrust normalized by surge amplitude and its phase shift with respect to the surge motion is shown in Fig. 4. Good agreement is noted between the experimental data and the numerical models, and results are in-line with those showcased by Bergua et al. during the OC6 Phase III project (Bergua and et. al., 2023). Interestingly, a linear relationship between normalized amplitude and frequency can be noted for all the numerical models up to 4 Hz. This indicates that the oscillation in thrust force is directly proportional to the relative velocity, as demonstrated in the following. Rotor thrust force can be written as:

$$F_x = \frac{1}{2}\rho C_t A U^2 \tag{3}$$

Where $C_t$ is the thrust coefficient, $A$ is the rotor area and $U$ is the incoming air speed. Assuming a small variation in inflow, the corresponding variation in thrust can be written as:

$$\Delta F_x \approx \rho C_t A U \Delta U \tag{4}$$

Equation 4 implicitly assumes $C_t = const.$, which is only valid in proximity of the chosen operating point. Assuming a sinusoidal variation in surge $x = x_0 * \sin(2\pi f t)$, the relative inflow to the rotor can be written as:

$$U = U_\infty + \dot{x} = U_\infty + 2\pi f x_0 * cos(2\pi f t) \tag{5}$$

Therefore, assuming steady inflow, the amplitude of velocity variation is due to the surge variation only and can be written as:

$$\Delta U = 2\pi f x_0 \tag{6}$$

If Eq. 6 is substituted into Eq. 4 and divided by the oscillation amplitude, the normalized amplitude is found to be proportional to the oscillation frequency:

$$\frac{\Delta F_x}{x_0} \approx 2U_\infty \pi \rho C_t A f \approx K f \tag{7}$$

Therefore, a linear relationship between normalized amplitude and surge frequency indicates quasi-steady behaviour. Deviations from such trend are an indication of non-linearities in the analysed models. It is important to note that deviations from the linear trend can be a result of unsteady aerodynamic effects but also a result of other model non linearities such as those present in the lift and drag coefficients. It is also important to keep in mind that the results discussed in the following are valid only for the operating condition that is being analysed; since the wind turbine is a non-linear system, the non-linearities



may be more or less evident depending on the specific operating condition. In this context, as shown in Fig. 4, the model analysed herein behaves quasi-steadily up to 4 Hz, and all the numerical models, including the simple BEM-based ones, are

able to correctly predict the thrust variation amplitude. This specific frequency is significant as it corresponds to an oscillation with a period of 6 seconds at full scale, which is at the upper range of a typical wave excitation frequency band. Thus, higher frequency oscillations can hardly be related to linear wave excitation. Although high-frequency oscillation may result from tower deformation, these motions are typically small and are not exclusive of FOWTs. Tension Leg Platforms (TLPs) could be the exception to this, as these platforms are designed with pitch natural frequencies above the wave excitation range, that

could be excited by non-linear forcing, resulting in pitch motion at high frequencies.

On the other hand, the influence of unsteady aerodynamic effects can be seen in the phase shift of the thrust oscillations even when the model behaves linearly. In fact, as shown in Eq. 4, in absence of unsteady effects, thrust oscillations are expected to be proportional to oscillations in relative velocity. Therefore, in the case of a harmonic surge excitation, rotor thrust is expected to lag behind the surge motion by 90°. The only model that follows the quasi-steady trend is BEM with static polars (BEM st).

When unsteady blade aerodynamic effects are included (dynamic polars), the thrust force phase shift is smaller than -90° and thus lags behind the relative velocity, as also noted by Mancini et al. (Mancini et al., 2020) and Bergua et al. (Bergua and et. al., 2023). This behaviour can be noted in the BEM and DBEM model trends in Fig. 4c and is also highlighted in the LLFVW and ALM models, despite them using static lift and drag tables. In fact, the phase lag is a consequence of the unsteady vorticity shed from the blade, as first explained by Theodorsen (Leishman, 2016), an effect that both the LLFVW and ALM models are

able to explicitly resolve.

Focusing back on the normalized thrust force in Fig. 4 (a), at 8 Hz, 12 Hz and 16 Hz, the LLFVW and ALM models deviate from the linear trend, differently from the BEM-based models, that continue to behave quasi-steadily despite the inclusion of unsteady blade aerodynamics in the BEM model and dynamic inflow models in the DBEM model. These results are consistent with those of Ribeiro et al. (Ribeiro et al., 2022, 2023), who found a similar trend in surge simulation of the UNAFLOW rotor

using a panel code. These results show how Blade Element based codes, if coupled to a higher order wake theory, are also able to capture this effect. Ribeiro et al. (Ribeiro et al., 2022) attribute the non-linear ΔFx variations to Theodorsen unsteady attached flow effects. More recently, in a WESC 2023 presentation (Schulz, 2023), Schulz attributed the dip in $\Delta F_X/A_{surge}$ that can be seen at 12 Hz to the returning wake effect, an aerodynamic phenomenon first noted by Loewy (Leishman, 2016). This effect is an extension to Theodorsen's theory that typically manifests in rotors and consist in non-linear variations in

aerodynamic forces on a plunging blade due to interaction with the vorticity shed, not only by the blade in examination, but also by the other rotor blades and the returning wake of the blade itself. In the current test case, this effect is most noticeable when the surge frequency is 12 Hz, as this corresponds to three times the revolution frequency, i.e., $f_{surge} = n_{bld} * f_{rot}$, where the effects of the shed vorticity from the three rotor blades are in-sync (Leishman, 2016). In fact, when increasing the oscillation frequency even further to 16 Hz, the ALM and LLFVW models continue to deviate from the linear frequency - normalized

amplitude trend (Fig. 14 (a)), although to a lesser extent in relative terms (Fig. 14 (b)). BEM models on the other hand, are unable to capture the returning wake effect, as they do not include the mutual interaction of shed vorticity from other rotor





blades, and do not show the dip in ΔFx/ΔAs that the higher order models manifest. These unsteady effects may be larger in a scaled-model test case as they grow larger as airfoil reduced frequency ($f_{ra} = \pi f * c/U_{rel}$) grows larger, and chord lengths are typically increased in scaled models to increase rotor thrust at low Reynolds numbers.

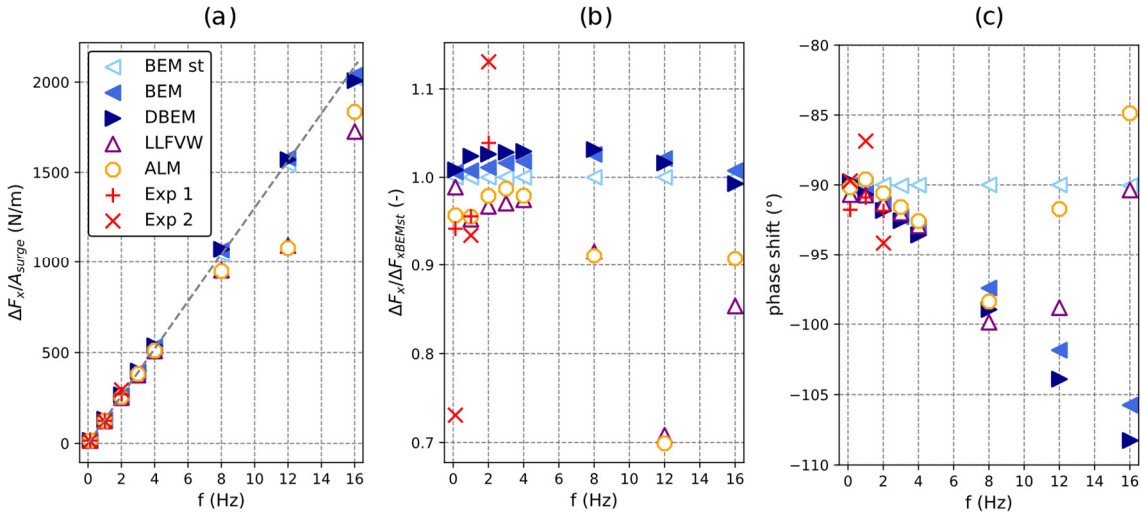

**Figure 4: Normalized rotor thrust as a function of surge oscillation frequency during tests with harmonic rotor surge motion. (a) Thrust amplitude normalized by surge amplitude, (b) thrust amplitude normalized by quasi-steady model predictions (BEM st) and (c) phase-shift of predicted rotor thrust with respect to surge motion. Filled indicators for models with dynamic polars, blank indicators for models with static polars.**

It is also important to note that dynamic effects may be more important in the case of larger rotors. This may be due to several factors. Firstly, as explained in Section 2.2, the model tests presented herein are scaled based on reduced frequency (Bayati et al., 2017a), which depends on rotor diameter, disproportionally affecting larger rotors. Therefore, assuming that reduced-frequency scaling is relevant for aerodynamic unsteady effects, these may occur at lower full-scale frequencies on larger rotors, such as on 15+ MW turbines. This is compounded in the context of returning wake effects, as larger rotors also feature lower rotational speeds. Finally, some platform concepts, such as Tension-Leg Platforms (TLPs), typically feature pitch natural frequencies in the 0.2-0.25 Hz range (Matha, 2009), above ordinary wave excitation. Phenomena such as non-linear wave forcing or 3P aerodynamic forcing could excite these resonance modes and induce oscillations and unsteady aerodynamic effects. Further research is needed to fully understand the implications of these phenomena.

At lower oscillation frequencies, differences in the aerodynamic forces predicted by the different wake models can be seen if harmonic oscillations in blade pitch and rotor speed are introduced with the surge oscillation. Results from 2-Hz frequency tests are shown in Fig. 5. When an oscillation in rotor speed is introduced (LC 2.16), wake dynamics cause an increase in the aerodynamic thrust amplitude, while when a blade-pitch oscillation is introduced (LC 2.17), aerodynamic thrust amplitude is lower for the LLFVW and DBEM models. On the other hand, very little differences in rotor thrust amplitude are noted in case of LC 2.12, where the same thrust oscillation as LCs 2.16 and 2.17 is imposed, but no blade-pitch or rotor speed oscillation is present. The different behaviour of the numerical models in LC 2.12, where no dynamic-inflow effects are apparent in the





thrust force amplitude, respect to LCs 2.16 and 2.17, where they are, is hard to explain given that axial induction along the blade span, a metric directly related to the "intensity" of the wake, varies to a similar degree in all three test cases (Fig. 5). Moreover, Fig. 6 highlights how in absolute terms, axial induction reaches higher values il LC 2.12. In this load case, axial induction is greater than the threshold of 0.4 in the outer 15% of the blade, where momentum theory is invalid and specific

high-induction corrections are applied to BEM-based solutions in AeroDyn. Moreover, while these results are in-line with results from participants to the OC6 phase III numerical experiment (Bergua and et. al., 2023), the lower amplitude of the thrust force in case of a blade-pitch oscillation appears counter-intuitive because over or under-shoots in rotor forces with respect to quasi-steady wake theories are noted in the case of blade pitch step-tests (Øye, 1991).

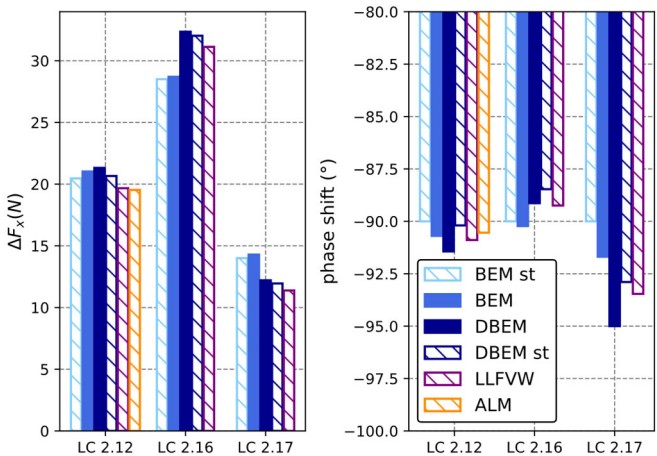

**Figure 5: Amplitude and phase shift of rotor force under 2 Hz harmonic surge motion (LC2.12), harmonic surge motion and rotor speed variation (LC2.16) and harmonic surge motion and blade pitch variation (LC2.17). Filled bars represent models with dynamic polars, banded bars models with static polars.**

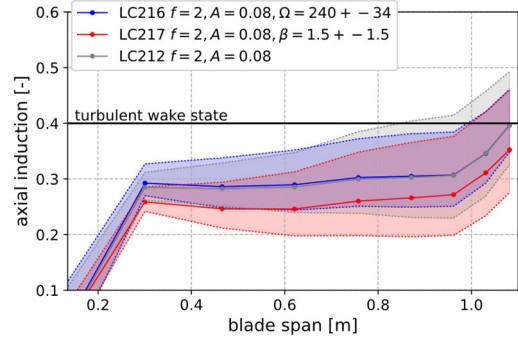

**Figure 6: mean (lines) and variation range (shaded areas) of axial induction as a function of blade span for blade one for the**
**LLFVW model. Axial induction is computed through Eq. 15 during post-processing.**



However, wake dynamics in the case of oscillatory tests such as those shown in LCs 2.12, 2.16 and 2.17 are arguably different from those occurring during a step-test. To illustrate this, step tests in surge velocity, blade pitch and rotor speed for the UNAFLOW rotor are shown in Figs. 6 and 7. In these tests, the rotor is operating at rated conditions, with a wind speed of

4.19 m/s. The magnitude of the blade-pitch, rotor speed and surge velocity steps are the same as the amplitude of the oscillations imposed in LCs 2.12, 2.16 and 2.17; 3° step in blade pitch, 2 m/s step in surge velocity and 72 rpm step in rotor speed. The duration of the steps is 0.2 seconds, which corresponds to half a cycle in the case of the LCs analysed in Fig 5.

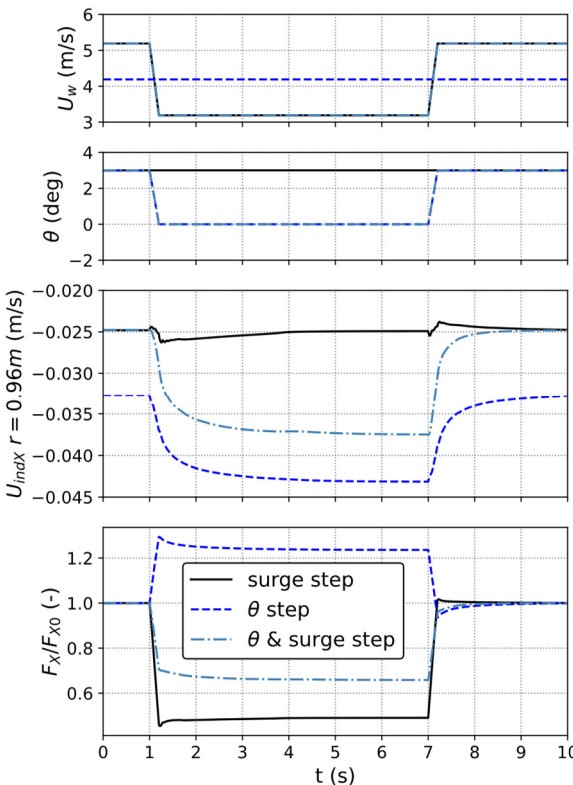

**Figure 7: Blade pitch and rotor surge step tests computed using LLFVW. Top to bottom: wind speed, blade pitch, axial induced velocity at 88% blade span and thrust force normalized by starting rotor thrust.**

As shown in Fig. 7, in the case of a simple surge velocity step, there is very little aerodynamic load under/over-shoot. This phenomenon was also noted by other authors (Schepers, 2022; Corniglion et al., 2022), that found little variations in induced velocity despite significant variations in axial induction. In particular, Corniglion et al. (Corniglion et al., 2022) ran similar analyses on the UNAFLOW rotor and attributed the lack of dynamic load over/under shoot to the fact that the step change in

surge velocity causes the blade tip-vortex spacing in the near wake to vary, partially compensating for the induction change along the blade. The magnitude of this cancellation effect depends on the rotor design and on the operating point in exam. The dynamic inflow effect, as noted and described by Snel and Schepers (Snel and Schepers, 1995), can be noted in the case of a



blade-pitch step, with rotor thrust taking up to 3.5 s to reach a steady state value. Interestingly, this effect is also present when a concurrent blade-pitch and surge velocity steps are introduced.

Similar considerations can be drawn in case of a rotor speed step test: dynamic load over/under shoot can be observed in the case of a rotor speed step and can be noted also when rotor speed and surge velocity steps are combined, despite this dynamic effect being present to a much smaller extent in the case of a surge velocity step.

Induced velocity timeseries in the outer part of the blade are also shown in Figs. 7 and 8. In-line with previous scientific literature, during both rotor speed and blade-pitch steps, variations in induced velocity are much greater than those recorded

during a surge velocity step. While not to the same extent, this consideration holds true for most of the blade. These results are not shown here for brevity and to avoid overlap with the work of Corniglion et al. (Corniglion et al., 2022), to which the reader is referred for further considerations.

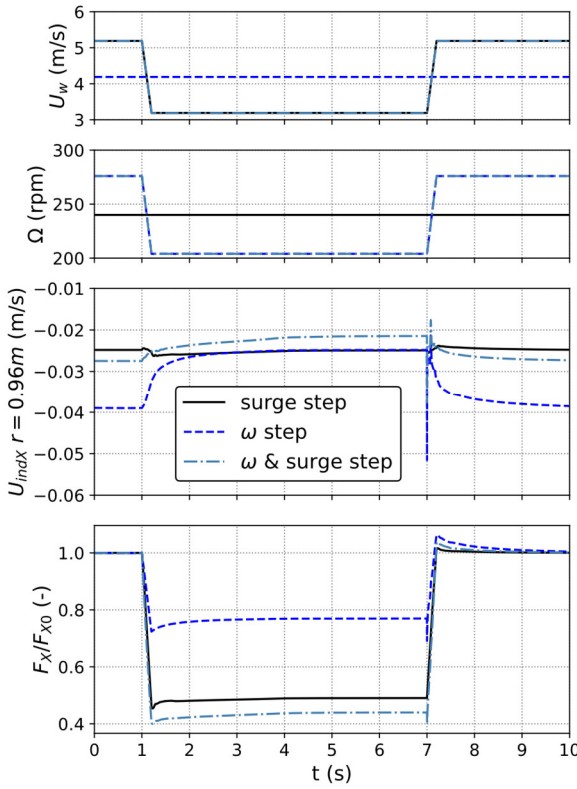

**Figure 8: Rotor speed and rotor surge step tests computed using LLFVW. Top to bottom: wind speed, rotor speed, axial induced**
**velocity at 88% blade span and thrust force normalized by starting rotor thrust.**

The timescales of the dynamic wake effect for this rotor at rated operating conditions can also be deduced from the data shown in Figs. 7 and 8, as thrust force requires 3-4 seconds to reach its steady state value. In the case of the oscillatory load cases examined in this study, an entire cycle lasts 0.5 s. Therefore, while wake physics remain the same, we argue that a different dynamic effect is being observed herein, and that a straightforward load overshoot when dynamic inflow effects are accounted





for cannot be expected. In this context, the fact that the simple dynamic wake model integrated in the tested DBEM model is
able to improve agreement in the prediction of aerodynamic force amplitude for an oscillatory test is a good indication that the
model is able to capture the general physics of the dynamic wake problem at hand. This is indeed very different form a step
test in rotor speed or blade pitch for which such dynamic wake models were developed and tuned.

For more insight on how dynamic wake effects influence aerodynamic forces, the spanwise amplitude and phase shifts of axial
induced velocity and relative axial velocity are shown in Fig. 9. Focus is placed on these parameters as they strongly influence
thrust force variations. In fact, the spanwise thrust force per unit span of each blade can be written according to blade-element
theory as:

$$F_x = \frac{1}{2}\rho c\, C_{ax}(\alpha, \vartheta, \varphi) U_{rel}^2 \tag{8}$$

Where $\alpha$ is the local angle of attach $\alpha = \tan^{-1}\left(\frac{U_{relY}}{U_{relX}}\right)$, $\vartheta$ is the blade pitch angle and $\varphi$ is the local twist angle. The relative
velocity can be decomposed into axial and tangential components, as radial components do not generate aerodynamic loading
in Blade-Element based models:

$$U_{rel}^2 = U_{relX}^2 + U_{relY}^2 \tag{9}$$

The axial and tangential velocity components can be written respectively as:

$$U_{relX,Y} = U_{\infty X,Y} + U_{indX,Y} - U_{stX,Y} \tag{10}$$

Considering that structural velocity is the same for all the compared wake models, that the wind speed is aligned with the rotor
($U_{\infty Y} = 0$), and in the hypothesis that tangential induced velocity is small ($U_{indY} \cong 0$), $U_{relY}$ can be considered to be
independent from the wake model, and focus can be put on axial relative velocity $U_{relX}$ only, that directly influences
aerodynamic thrust (Eq. 8).

In Fig. 9, the spanwise amplitude and phase shift with respect to the imposed surge motion of the relative blade axial velocity
are shown on the bottom two rows. In the top rows, the phase shift with respect to the imposed surge motion and variation
amplitude as a function of blade span of the induced velocity is shown. The columns correspond to LCs 2.12, 2.16 and 2.17.
If the spanwise amplitudes of induced velocity variation for the three load cases are compared to each other in Figs. 9 (b), (f)
and (l), similar $\Delta U_{indX}$ can be noted for all three LCs. This indicates that the different model behaviour observed in the three
LCs – i.e. lack of differences between dynamic-wake models and quasi-steady ones in LC 2.12, but different thrust force
amplitudes for models with and without dynamic-wake in LC 2.16 and 2.17 – are not caused by lower variation in $U_{indX}$, as
is the case for step-tests (Figs. 6 and 7). Focusing on LC 2.12, with respect to the BEM model, which doesn't include wake
dynamics, the DBEM, LLFVW and ALM models show significant differences in relative velocity. All three models predict
lower relative velocity oscillations in the outer span of the blade but higher oscillations in the inner part, leading to the
differences between them cancelling out when the global rotor thrust is considered. Given that the structural windward velocity
and the wind speed are the same for all the blade stations and all numerical models, as surge motion is being analysed, this is





a result of differences in induced velocity. In this regard, the induced velocity amplitude along the blade span is lower on the entire blade for the models that include dynamic induction. This is consistent in all three LCs analysed in Fig. 9. The lower variations in induced velocity lead to higher or lower relative velocity amplitudes depending on how the former combine with oscillations in structural velocity (Eq. 10). To this end, it is useful to focus on the phase-shift of the induced velocity (Fig 8 a, e, i). Focusing on BEM results, which do not include phase lags and are thus easier to interpret, the phase-shift of the induced velocity shifts from -90° in the outer part of the blade to +90° in the inner part of the blade. On the other hand, the structural velocity lags behind surge motion and is thus shifted -90° for all blade stations.




**Figure 9: Phase-shift (top) and amplitude (middle) of induced velocity and amplitude of relative velocity (bottom). LC 2.12 (a-d), LC 2.16 – harmonic rotor speed variation (e-h) - and LC 2.17 – harmonic blade pitch variation (i-n).**






Therefore, if induce velocity and structural velocity variations are in-phase, their combination increases the variation in axial velocity, while if the phase shift of the induced velocity is +90° the two signals are in phase-opposition and thus their combination leads to lower relative velocity. Therefore, in the case of LC 2.12, this phenomenon has the effect of augmenting

the velocity oscillations in the outer part of the blade and diminishing them in the inner part. Despite the models that include wake dynamics having different phase shifts, this reasoning can be applied to the latter models as well. Therefore, because the models that account for dynamic induction have lower induced velocity amplitudes, they show lower variations in induced velocity in the outer part of the blade and higher ones in inner part with respect to BEM. The way structural and induced velocities variations combine is also the reason differences between the numerical models emerge in LC 2.16 (Fig. 9 e-h) and

2.17 (Fig. 9 i-n). Similarly to LC 2.12, the wake dynamics act like a filter and reduce the amplitude of the induced velocity variations along the span. In LC 2.16 because the phase-shift of $U_{indX}$ is positive, more variation in relative velocity for DBEM and LLFVW can be noted. On the other hand, because the $U_{indX}$ phase-shift is negative along most of the blade span in case of LC 2.17 (Fig 8 i), and thus variations in induced velocity are in-phase with the structural velocity, leading to less induced velocity variation and less aerodynamic thrust variation in DBEM and LLFVW.

Finally, it is interesting to investigate why the $U_{indX}$ phase shift may assume positive or negative values depending on the spanwise location and test case. These can be explained by considering the induced velocity as:

$$U_{indX} = (U_\infty - U_{strX}) * a \qquad (11)$$

Where a is the induction factor for each radial station. In the case of more advanced theories such as LLFVW and ALM, where a is not computed through a momentum balance, this factor is used to summarize the effect of the wake on each radial station.

It is clear from Eq. 11 that variations in $U_{indX}$ are a result of the combined variations of $U_{strX}$ and $a$. For the UNAFLOW rotor, the two quantities combine in different ways along the span, leading to the observed behaviour il LC 2.12. In summary, dynamic wake effects act like a filter on the induced velocity and reduce its variation in all three LCs 2.12, 2.16 and 2.17. In the case of simple surge motion (LC 2.12), no dynamic inflow effects on the loads are observed due to this effect leading to higher load amplitudes in the outer part of the blade that are compensated by lower amplitudes in the inner part.

**3.2 Cut-in wind speed**

BEM-based models are called to perform reliably in even more challenging conditions than those examined in section 3.1, despite tests carried out in this section highlighting some differences between the aerodynamic theories. In fact, as discussed in Section 2, BEM models are challenged when rotor loading is high and specific high-induction empirical corrections need to be employed. In this view, rotor loading generated from the combination of surge oscillation and inflow conditions tested in

section 3.1 do not push BEM models to their limit (Fig. 6). On the other hand, at lower wind speeds, modern wind turbine rotors are typically highly loaded, thus motivating the examination of these conditions. Moreover, we argue that rotor-wake interaction is most likely in these operating conditions for a floating rotor. In fact, interaction of the rotor with its own wake, as in the rotor blades moving in and out of their own wake, is most likely when:




1. downstream wake convection velocity is low. This can be achieved through a combination of low incoming wind
speed and high rotor loading, that leads to high axial induction and thus lower velocity downstream of the rotor plane.
2. fore-aft rotor velocity of the blade elements is high. In case of harmonic oscillatory tests such as the ones performed
        herein, this can be achieved through a combination of high amplitude and high frequency of the oscillations.

Based on these considerations, two additional LCs are evaluated, namely LC 3.26 and 3.27 as shown in Table 2. In these LCs,
the turbine is operating at a wind speed of 5 m/s, close to the cut-in wind speed. In these inflow conditions, many variable
speed rotors, including the DTU 10MW, upon which the UNAFLOW rotor is based upon, operate at minimum rotor speed (to
avoid tower resonance) and are forced to run at a high tip speed ratio (TSR). This leads to high rotor loading, which combined
with the low incoming wind speed, meets the criteria highlighted in point n°1. The tests are performed with a forced floater-
pitch oscillation of 1° and 2° at a full-scale equivalent frequency of 0.1 Hz, which is commonly in the wave excitation range.
A floater-pitch oscillation is favoured over a surge oscillation as the former can leverage the full turbine height, inducing larger
fore-aft oscillations in the upper part of the rotor. As shown in the following sections, these conditions indeed lead to rotor-
wake interaction on the selected test case. These imposed oscillations on the model-scale test case treated herein can be put
into context and compared to wave conditions that would lead to similar fore-aft velocities on full-scale research FOWTs. The
maximum blade tip fore-aft velocity induced by a pitch oscillation, when the blade is pointing upwards can be written as:

$$V_{tip} = 2\pi f A(l_{twr} + l_{bld}) \tag{12}$$

Where $l_{twr}$ and $l_{bld}$ are the tower and blade lengths, and $z_{CoG}$ is the position of the FOWT center of gravity below the sea
water level. The center of rotation is assumed to correspond, as a first approximation, with the center of floatation of the
system. Although the position of the center of ration is influenced by many factors, such as external forcing, and position of
center of gravity and center of buoyancy, this decision is supported by recent research (Patryniak et al., 2023) indicating a
strong alignment between the center of rotation and the center of flotation, particularly evident under higher wave excitation
frequencies. By inverting Eq. 10, the required oscillation amplitude to reach an equivalent $V_{tip}$ can be expressed as:

$$A(f) = V_{tip}/2\pi f(l_{twr} + l_{bld}) \tag{13}$$

Finally, the height of a regular wave required to reach an oscillation with amplitude $A(f)$ can be expressed as:

$$H_w(f) = \frac{2A(f)}{RAO(f)} \tag{14}$$

Since Response Amplitude Operators (RAOs) typically relate two-sided wave heights to two-sided oscillation amplitudes, the
required amplitude is doubled. The resulting wave heights derived for common research turbines are shown in Fig. 10 for pitch
oscillations (a) and surge oscillations (b), in which case Eq. 12 is replaced by $V_{tip} = 2\pi f A$. RAOs are extracted from the plots
in (Ramachandran et al., 2013) for the NREL 5MW OC3, and from (Papi and Bianchini, 2022) for the NREL 5MW OC4 and
IEA 15MW models.





**Table 3: Dimensional characteristics of floating reference turbines.**

| Turbine | NREL 5MW OC4 | NREL 5MW OC3 | IEA 15MW Semi |
|---|---|---|---|
| blade length [m] | 63 | 63 | 120 |
| tower height [m] | 90 | 90 | 150 |
| CoG below SWL [m] | -13.5 | -89.9 | -14.94 |

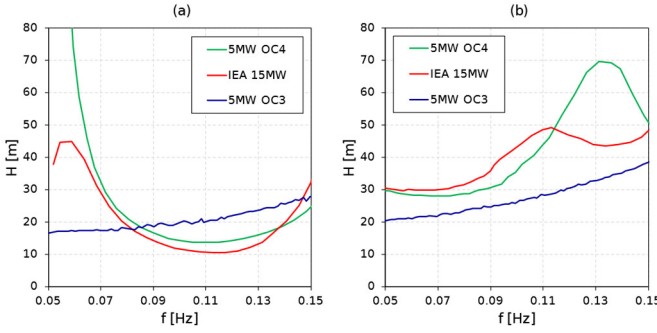

**Figure 10: Required two-sided wave height for blade tip velocity to reach value reached in LC 3.26 (4.357 m/s at full scale) through pitch motion (a) and through surge motion (b).**


When Fig. 10 (a) and Fig. 10 (b) are compared, as expected much higher wave heights are required to reach the specified fore-aft velocity in case of a surge motion. Focusing on pitch motion (Fig. 10 (a)), wave heights of approximately 10 to 13 meters are required between 0.09 and 0.13 Hz to reach a rotor apex fore-aft velocity equivalent to LC 3.26 at full scale. If we consider LC 3.27, where oscillation amplitude is halved, half the wave height is required due to the linear nature of RAOs.

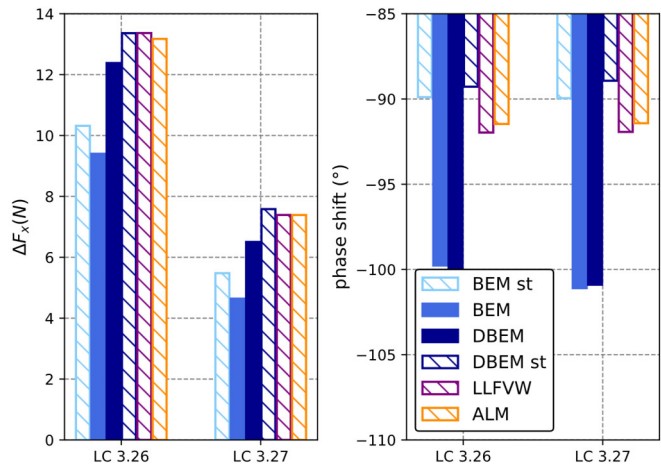


**Figure 11: Amplitude and phase shift of thrust force at cut-in wind conditions. Filled bars represent models with dynamic polars, banded bars models with static polars.**





The predicted amplitude and phase shift of rotor thrust in these conditions (LCs 3.26 and 3.27) is shown in Fig. 11. The thrust force oscillations are again summarized by reporting oscillation amplitude and phase-shift values, as the imposed oscillations
cause variations in aerodynamic forces mostly at the oscillation frequency itself. Significant underprediction of thrust force oscillation can be noted for the BEM model. On the other hand, DBEM, LLFVW and ALM are all very close in magnitude, indicating that, once again, a medium fidelity aerodynamic theory such as LLFVW is very close to a high-fidelity wake model such as ALM in the prediction of rotor forces, and that the dynamic inflow model implemented in DBEM performs well in the prediction of global rotor loads in these conditions. Differently from LCs 2.12, 2.16 and 2.17 (Fig. 5) significant differences
in both amplitude and phase shift can be seen when dynamic polars are used rather than static ones.

To investigate this more in detail, spanwise distributions of axial induction, relative axial velocity and out-of-plane aerodynamic force per unit length in LC 3.26 are shown in Fig. 12. Spanwise distributions are shown for blades one and two because the revolution frequency is synchronized with the oscillation frequency. Therefore, as pitch motion introduces asymmetric axial velocity distributions, spanwise quantities are different on the three blades. On both blades shown in Fig 12
(a) and (b), axial induction exceeds the value of 0.4, where momentum theory is considered invalid. Moreover, axial induction reaches values above unity in the outer 60% blade span for the LLFVW and ALM models on blade one (Fig. 12 (b)).

In case of the LLFVW and ALM models, which do not explicitly solve for axial induction, this parameter is computed in a post-processing phase by inverting Eq. 1 as:

$$a = 1 - \frac{U_{rel}^i}{U_{wind}^i + U_{str}^i} \qquad (15)$$

As discussed in Section 2, AeroDyn includes structural velocity in the momentum side and in the blade-element side of the momentum balance and is consistent with Eq. 15. A direct consequence of this is that induction factors above unity are directly linked to flow reversal on the rotor blades. This can be noted by comparing Fig. 12 (a) and Fig. 12 (c), where negative inflow on the blades can be seen in the outer 40% of blade one where axial induction is above unity for LLFVW and ALM. Fig. 12 (c) also confirms that rotor-wake interaction, as hypothesized in section 3.2, does indeed occur in LC 3.26. However, as shown
in Fig. 12 (b), no flow reversal occurs on blade two. Analogous considerations can be drawn for blade three, not shown for brevity. Flow reversal can be noted especially in the LLFVW and ALM models in Fig. 12 (c), which also reach the highest axial induction values (Fig. 12 (a)). Negative relative inflow near the tip of blade one is also predicted by DBEM, although to a lesser extent. On the other hand, BEM does not predict flow reversal on the rotor (Fig. 12 (c)) and axial induction values always remain below unity (Fig. 12 (a)). The influence of wake dynamics is apparent in these test conditions: BEM greatly
underestimates the amplitude of relative axial velocity on both blades. The inclusion of dynamic wake effects through Øye's dynamic inflow model improves agreement with higher order theories greatly, falling somewhat short of the latter only in the outer part of blade one, where the model appears to be pushed to the limit. Spanwise axial force per unit length along blades one and two are compared in Fig. 12 (e, f). Analogously to the relative velocity, DBEM is able to improve the prediction of the variation amplitude of axial force along the span for blade two. It also improves predictions with respect to BEM on blade



one but only in the inner 60% of the span. Indeed, DBEM's dynamic induction correction brings this model's results more in line with higher order theories in the inner part of the blade, where induction factors are lower, but fails to do so in the outer part (Fig. 12 (b)). Despite this, the combination of aerodynamic force variation on the three blades leads to thrust force oscillations very similar to higher order theories, as shown in Fig. 11. On the other hand, LLFVW and ALM are again very close in both mean values and variation amplitudes in all the quantities in Fig. 12. In summary, the ability to correctly predict

spanwise force variations in the codes is closely linked to a correct prediction of relative inflow velocity and axial induction.

**Figure 12: mean (lines) and variation range (shaded areas) of axial induction as a function of blade span for blade one (a) and two (b), relative axial velocity along blade span for blade one (c) and two (d) and axial force per unit length along blade span for blade one (e) and two (f). Values computed using LLFVW model.**



### 3.3 Wake states

Axial induction values are often linked to the wake state the rotor is operating in. In particular, axial induction values above unity conventionally indicate that the rotor is operating in *vortex ring state* (VRS), while values between 0.4 and 1 are linked

to the *turbulent wake state* (TWS). In both cases, momentum theory is invalid and specific empirical models are typically applied in BEM-based codes. Moreover, many codes use axial induction as a threshold to switch between submodels that have been developed over time to model specific conditions such as TWS or VRS. For instance, this is the case for AeroDyn (Ning et al., 2015) and TNO's code AeroModule (Mancini et al., 2022), despite both codes not explicitly solving for axial induction. We argue that, from a theoretical standpoint, such a direct link between axial induction and wake state can only be assumed in

the case of a static rotor operating in steady-state inflow conditions, where the momentum balance is applied in an inertial reference frame. In case of a floating rotor, the high induction factors observed in Fig. 12 (a) are simply a consequence of the momentum balance being applied in a non-inertial frame and are due to the unsteady rotor-relative velocities. To demonstrate this point, the axial velocity field in the rotor wake during LC 3.26 shown in Fig. 13 can be observed. The velocity in the rotor reference frame is derived as follows; the structural velocity in the axial direction due to pitch motion can be written as:

$$U_{strX}(t) = V_t(t)\cos(\theta(t) + \phi) \tag{16}$$

where $\phi$ is the geometric angle deriving from the combination of rotor ovehang and height above the oscillation point, $R$ is the distance from the oscillation center, $R_z$ being the height and $R_x$ the rotor overhang, $V_t(t)$ is the rotational velocity due to pitch motion and $\theta(t)$ is the instantanous pitch angle:

$$\phi = \tan^{-1}\left(\frac{R_x}{R_z}\right)$$

$$R = \sqrt{R_x^2 + R_z^2} \tag{17}$$

$$V_t(t) = 2\pi f A \cos(2\pi f t)$$

$$\theta(t) = A\sin(2\pi f t)$$

With the structural velocity defined as in Eq. 16, this component can be subtracted from the velocity field to obtain the axial flow velocity in the rotor reference frame. The result of this operation is shown in Fig. 13. For each time instant, the axial

velocity in the fixed inertial reference frame is shown on the left, while the axial velocity in the rotor reference frame is shown on the right. Large degrees of flow reversal in the upper part of the flow field can be seen in the relative reference frame. However, in the absolute frame, no such reversal occurs. Significant degrees of wake expansion can be seen, indicating that the turbine is likely operating in turbulent wake state (TWS), a condition that is compatible with the high mean rotor loading of this operating point. Despite this, in the static reference frame, the wake appears to be in line with tipycal windmill wakes,

and flow structures typical of Vortex Ring State (VRS) or Propeller Brake State (PBS) cannot be observed.



This is in line with the observations of Ferreira et al. (Ferreira et al., 2022), that argue that TWS and VRS are a property of the stramtube rather than the rotor. Therefore, the high induction factors observed in Fig. 12 (a), are a consequence of rotor motion leading lo localized flow reversal on the rotor rather than an indication of TWS or VRS.

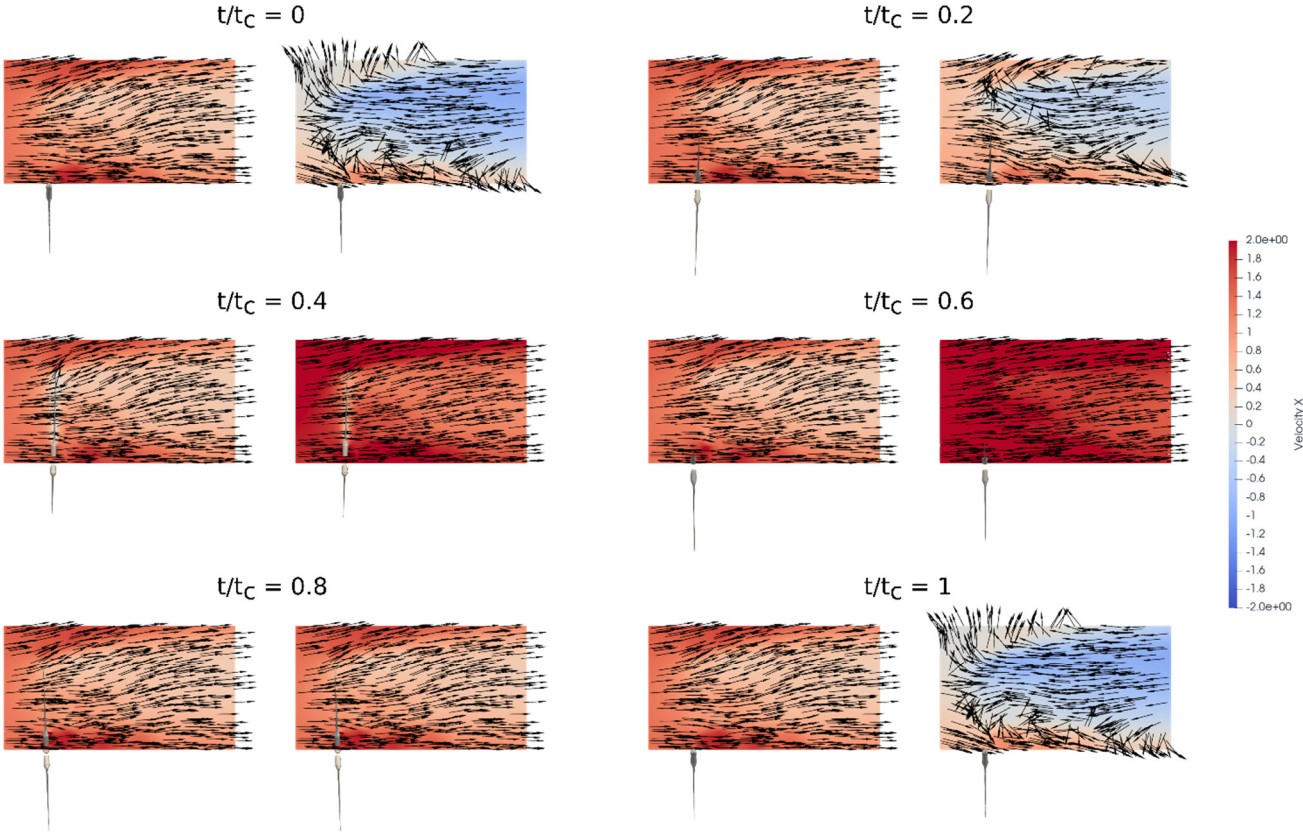

**Figure 13: Velocity field in the near wake of the UNAFLOW rotor during an oscillation cycle in LC 3.26. Colormap is relative to axial (X) velocity, vectors indicate in-plane direction of the fluid. Absolute velocity field in inertial reference frame (left) and relative velocity field in rotor reference frame obtained with Eq. 16 (right). Only upper half of mid-wake is shown as relative inflow variations are largest.**

Despite this, applying the BEM-based model in the rotor reference frame, as if it were in TWS or VRS, yields good results if

a dynamic inflow model (DBEM) is included in the analysis, as shown in Fig. 10, although falling somewhat short of higher-order models in the prediction of induced velocitiy oscillations in the outer part of blade #1, which encounters the most variations in relative inflow (Fig. 12). Moreover, this strategy has proven to work well also in nominal operating conditions at rated wind speed (Figs. 4, 5). Boorsma and Carboni (Boorsma and Caboni, 2020) came to a similar conclusion when comparing lifting-line and BEM results of surge oscillation tests of the UNAFLOW rotor; including platform surge motion in the

momentum balance as "apparent wind" lead to improved agreement with LLFVW. The same was noted by Mancini et al.



(Mancini et al., 2022) in a subsequent work, although it must be pointed out that, as noted by the authors, applying a momentum balance in the rotor reference frame, or even in unsteady conditions, is theoretically incorrect, and thus such models fully qualify as engineering, empirically derived, approaches.

In this view, the results of this study confirm that treating structural velocity variations the same way as inflow velocity varitions allows for more effective prediction of dynamic aerodynamic loading in BEM-based models, especially in cases where variations in structural velocity are smaller compared to the mean relative velocity on the blades, such as LCs 2.12, 2.16 and 2.17 that are shown in Fig. 5.

## 4 Conclusions

In this study, a critical analysis on the applicability and accuracy of BEM-based aerodynamic models in complex flow conditions typical of floating wind turbines is proposed. To this end, the aerodynamic forces resulting from forced surge and pitch oscillation tests of the UNAFLOW rotor are used. In an attempt to find the real limits of BEM, we also went beyond the UNAFLOW tests by analysing oscillations with higher frequency and additional load cases in low wind speeds. The amplitude and frequency – 0.1 Hz at full scale – of these additional tests was selected as to be representative of extreme wave-induced oscillations.

Results have shown how in rated wind speed conditions, all models are able to capture the amplitude of the thrust force oscillations. In these conditions, there is a linear relation between the normalized amplitude and oscillation frequency. At high oscillation frequencies, the LLFVW and ALM models stop behaving linearly. This effect is ultimately linked to unsteady airfoil effects, namely Lowry's returning wake problem. BEM and DBEM results are unable to capture this behaviour, despite including unsteady blade aerodynamics, as they do not model returning wake effects. The frequency at which these aerodynamic phenomena are most apparent depends on the specific rotor design. In fact, discrepancy between BEM results and higher-order wake theories is largest when the oscillation frequency is three times the revolution frequency – 12 Hz at model scale – but are also apparent at 8 Hz, a 33% reduction with respect to the 3P frequency.

Analysing the results of oscillating tests at 2 Hz with blade pitch and rotor speed oscillations, namely LCs 2.12, 2.16 and 2.17, we have found that DBEM is able to predict thrust force variations that are in line with higher fidelity models such as LLFVW and ALM. On the other hand, in line with the findings of previous studies (Bergua and et. al., 2023), simple BEM theory falls short of the other model predictions when blade pitch and rotor speed variations are introduced with the surge oscillations (LC 2.16 and 2.17). In addition, we have shown that the use of a model that can include dynamic inflow effects, either directly through model solution or empirically, lowers the variations in induced velocity in all three LCs. In other words, unsteady wake dynamics (a.k.a: dynamic inflow) effectively act like a filter on induced velocities, diminishing their amplitude even in the case of surge oscillations only. The lack of a difference between BEM and higher order theories in the case of simple surge oscillations (LC 2.12) is not due to a lack of unsteady aerodynamic effects in this case, but rather by the way variations in induced velocity and structural velocity combine along the blade span, which explains the rotor thrust overshoot noted during



rotor speed variations, the undershoot during blade pitch variation and the substantial equivalency between aerodynamic theories in the case of surge variation only. It is also important to note that the amplitude of the induced velocity variation
along the blade span in the three examined load cases is comparable. This differs to what is observed during surge velocity step tests, where, similarly to LC 2.12, no dynamic inflow effects are observed. In this case, however, the absence of such effects is mostly attributed to the small variations in induced velocities that can be observed in the tests, in turn caused by the change in tip-vortex spacing partially compensating for the change in axial induction factors.

In lower wind speeds, near cut-in wind speed where rotor loading is high, DBEM is able to predict similar results to LLFVW
and ALM and it is only when considering spanwise quantities along the blades that some difference with respect to higher order theories emerge. From this point of view, treating structural velocities in the momentum balance the same way wind velocities are treated, as is done in this study, seems to be effective. On the other hand, applying the momentum balance in the rotor reference frame, as this method implies, means that induction factors can reach very high values during specific transient events. This does not necessarily imply that the rotor enters TWS or VRS, where momentum theory is invalid, as no flow
reversal can be seen in the rotor wake despite induction factors exceeding unity. Flow reversal can be instead seen across the rotor plane, as this is due to the rotor structural velocity rather than the induced velocity.

In conclusion, dynamic inflow effects are present even in surge tests despite there being little to no difference in the predicted global aerodynamic thrust variation. Similar to the observations of other authors, performing the momentum balance in the rotor relative reference frame, effectively treating the structural velocity variation as a variation in inflow, despite being
theoretically incorrect, is practically effective. In fact, once augmented with a dynamic inflow model, DBEM is able to predict aerodynamic thrust variation in most of the analysed cases and still represents a valuable industrial tool also in the case of emerging FOWT technology. Lastly, this work is carried out on a scaled 10 MW rotor. The outcomes of the analysis may be more significant in the context of ever larger rotors that are being prototyped and built, which may be more influenced by dynamic inflow effects, as wake reduced frequency depends on the rotor diameter. Returning wake effects may also be more
likely to arise since larger rotors typically feature lower revolution speeds. In addition, as these dynamic effects are more prominent at high oscillation frequencies, TLP platforms, which feature high pitch natural frequencies, could be affected by resonance and thus induce unsteady aerodynamic effects that may not be appropriately captured by BEM models. More research is therefore needed in this regard.

**Author Contributions** This study represents one of the main outcomes of F. Papi's PhD research program. He conceptualized
the study and the additional tests beyond the UNAFLOW experiments, reviewed existing work, performed the numerical simulations, analysed the results, and wrote the first draft. A. Bianchini provided guidance in the PhD research project, discussed the idea and the results, contributed to the analysis of results, and reviewed the manuscript. A. Robertson provided support for the discussion of the implications of the main findings in the field of floating offshore wind energy and reviewed



the manuscript. J. Jonkman provided guidance in the analysis of NREL simulation tools and collaborated to the discussion of
the results; he also reviewed the manuscript.

**Competing interests** At least one of the (co-)authors is a member of the editorial board of Wind Energy Science.

**Acknowledgements** The authors wish to thank L. Pagamonci from the University of Florence for his support in running the
ALM simulations. Thanks are also due to Prof. G. Ferrara of the University of Florence for his guidance during F. Papi's PhD
program.

**Data availability** All data presented in the study are openly available upon request to the contact authors.

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
