# Peer review of "Going beyond BEM with BEM: an insight into dynamic inflow effects on floating wind turbines"

_Wind Energy Science, 2023_

## Author Comment (AC2)

**Going Beyond BEM with BEM: an Insight into Dynamic Inflow Effects on Floating Wind Turbines**

F. Papi, J. Jonkman, A. Robertson, A. Bianchini

Dear Reviewers, dear Editor,

Thank you for your time managing and reviewing our work and for your feedback. Based on the Reviewers' suggestions, we have done our best to improve the paper.

We have provided detailed answers to your comments below, in blue colored text for your convenience.

Best regards,

F. Papi, J. Jonkman,
A. Robertson, A. Bianchini

ooo   ooo   ooo

**Reviewer #1 (anonymous) comments:**

Dear authors,

thank you very much for the very well elaborated work. It is rare to have a submitted paper to be so well prepared on first submission. Thus, I only have a few very minor aspects to improve:

1. Please make sure, that all variables in the equations are explained at least once. Even if they seem very clear, in other papers, someone might use the exact same variable for something different, which also seems clear. To avoid this confusion, please define once. In most cases this is done, but not in all. So check.

Thank you for pointing this out. We have added explanation of the parameters used in the equations where missing, such as in equations 1 and 2, 8 and 9 and 13.

2. On line 303 there seems to be a typo before "LC 2.12"

Thank you again for pointing this out. Corrected.

3. In the paragraph from line 317 on, please use one or two sentences to explain first what step tests are and what they are used for before you go into the discussion.

As the Reviewer pointed out, not everyone may be familiar with this sort of test. We have added the explanation as requested (Lines 343-345)

4. The sentence starting in line 320 is not really good and hard to read. Please check it.

We have split the paragraph in two (L345-348). Hopefully it is clearer now.

5. On line 331 "and on the operating point in exam." What is exam?

This phrase generically refers to the fact that the mentioned cancellation effect depends on rotor design and operating point that is being considered. We have rephrased: "The magnitude of this cancellation effect depends on the rotor design and on the operating point under consideration". (L358)

6. In line 340 the sentence "While not to the same extend, this consideration holds true for most of the blade." is not very specific and clear. Either leave it away or get or precise, I would say.

The Reviewer is right. The change in induced velocity is greater for a step change in blade pitch and rotor speed respect to that recorded for a step change in surge velocity in the outer 50% of the blade. In the inner 20%, the changes in induced velocity are small for all step tests, while from approximately 20 to 50% of the blade, changes in induced velocity are greater for surge step tests. We have changed the paper to be more specific in this regard. (L373)

7. In line 352 "This is indeed very different form a step test", most likely it is "from a step test", isn't it?

Corrected, thank you!

8. On line 416: "BEM-based models are called to perform reliably in …", I would always say: "BEM-based models are said to perform reliably in …". What do you think?

The Reviewer has a point. On second thought this phrase does not flow very well. We meant to highlight that BEM models need to perform well also in other conditions. We changed the phrase to: "BEM-based models need to perform reliably in even more challenging conditions than those [....]" (L449)

9. On line 539 you write about windmill wakes – do you really mean windmills?

We were referring generically to "windmill" as a rotating device that extracts energy from the flow. "Wind turbines" is probably more appropriate and avoids confusion. We have changed the text to reflect this. (L586)

Otherwise, the paper opens a lot of room for specific aerodynamic discussion, which most likely cannot be finalized in one single paper. Thus, I find it good this way. But there are still open questions also coming from this paper, which will most likely need to be answered in the further discussions. I'm looking forward to it.

---

## Author Comment (AC3)

**Going Beyond BEM with BEM: an Insight into Dynamic Inflow Effects on Floating Wind Turbines**

F. Papi, J. Jonkman, A. Robertson, A. Bianchini

Dear Reviewers, dear Editor,

Thank you for your time managing and reviewing our work and for your feedback. Based on the Reviewers' suggestions, we have done our best to improve the paper.

We have provided detailed answers to your comments below, in blue colored text for your convenience.

Best regards,

F. Papi, J. Jonkman,
A. Robertson, A. Bianchini

ooo   ooo   ooo

**Reviewer #2 (M.O.L. Hansen) comments:**

The topic of investigating the performance of BEM codes for FOWT is very interesting and of big practical importance, but not very new. A very similar study was made 10 years ago by

J.B. de Vaal et al., Effect of wind turbine surge motion on rotor thrust and induced velocity, Wind Energy (2012)

We thank the Reviewer for pointing out the importance of this topic. We indeed did not cite the work that the reviewer pointed out in our manuscript. This was only on oversight from our side, and the work of J.B. de Vaal et al., that we were aware of, is now included in the initial discussion phase. Despite the existing body of scientific literature on the subject, we believe that there is still widespread lack of consensus surrounding the topic of FOWT aerodynamics. While the conclusions of this work are in many ways similar to those of J.B. de Vaal et al.'s work, we believe this is relevant as they are reached for a different rotor design, validated by more recent experiments and higher order models such as ALM and LLFVW, and also account for pitch motion in addition to surge motion.

The conclusion in the submitted paper is reconfirmed, that a BEM code with a proper dynamic inflow model and an empirical Glauert model for high thrust coefficients used on FOWTs performs quite well for the exposed structural oscillations (amplitudes and frequencies). It would be really nice, if the expected range of frequencies and amplitudes for the DTU 10MW rotor (pitch and surge) was included in the paper for different foundation types. This will also show if going above possible frequencies in the PoliMi tunnel is of practical importance for a real FOWT.

We thank the reviewer for this comment. For semi-sub and spar platforms natural frequencies in pitch and surge are around 0.005-0.05 Hz. Wave excitation range is typically 0.05-0.5 Hz (with 0.08-0.2 Hz being most common). Tension Leg Platforms (TLPs), on the other hand, usually feature higher natural frequencies in pitch, roll and heave, approximately in the 0.5 to 5 Hz range, although amplitudes are

typically small. Assuming reduced-frequency scaling is appropriate for this kind of aerodynamic phenomenon, it would make sense to go above the PoliMi test frequencies for this rotor. In fact, while the experimental apparatus is limited to approximately 2Hz, going up to 4Hz at model scale is well within the wave excitation range at full scale. It's important to note that reduced frequency is computed based on rotor diameter and wind speed as $f_r = \frac{fD}{U}$, therefore the appropriate frequency range at model scale varies based on working condition (wind speed) and turbine diameter, both at full scale and model scale. We have added a comment regarding the typical frequency ranges of utility scale FOWTs at lines 228-232.

The paper discusses different ways reported in the literature of how to treat the momentum equation in case of a dynamically oscillating wind turbine rotor. In basic fluid mechanics the conservation of momentum is used to determine an unknown force by keeping track of the total momentum deficit out of a control volume and including an inertia term in case of unsteadiness. The mean position of the turbine is not moving, so the control volume and velocities at the boundaries should be in a fixed frame of reference and not include the velocities of the rotor. These should in my opinion only be used when evaluating the angles of attack for the blade elements.

The Reviewer has raised an interesting point. From our understanding, this is somewhat of an open debate. We attempted to convey this in the introduction of this work. The Reviewer is suggesting, if we interpret correctly, that the structural velocity should be included only in the "Blade Element" part of the momentum balance and not in the "momentum" part. AeroDyn does not make this assumption, although the topic has been debated internally at NREL many times. Instead, structural velocity is included in both sides of the BEM balance in AeroDyn. As the Reviewer will agree, if no inertia terms are considered, however, including the structural velocity only in the "Blade Element" part of the BEM equations still violates the momentum balance. In fact, if the relative velocity at the actuator varies, axial induction will also vary, and thus the wake velocity is time-dependent, violating the steady-state assumption in which the momentum balance is formulated. Simulations and experiments have indeed shown that velocity measured downstream a surging rotor varies with time. (Cioni et. Al., 2023). Moreover, Boorsma and Caboni (Boorsma and Caboni, 2020) and Mancini (Mancini et al., 2022) have shown how including the velocity due to actuator motion in both sides of the momentum balance improves agreement with respect to LLFVW simulation that include it only in the "blade element" part. We have not yet encountered a "BEM" model that includes inertia terms in the momentum formulation. What we have observed throughout this work, is that in most realistic conditions, the BEM formulation implemented within AeroDyn, which is common to many wind turbine simulation codes, works well, despite its formal inconsistencies. This motivates the title of the paper "Going Beyond BEM with BEM". As the reviewer pointed out, we believe this topic is of practical importance, as it may help FOWT modelers make an informed decision on the modelling theories they use to approach such a problem. The introduction has been modified to reflect this discussion.

On page 5 and 6 the LLFVW is described as a dynamic vortex model where the vorticity is shed as vortex rings. Note that the coefficients in the Øye Dynamic Inflow model are actually calibrated from a similar dynamic ring vortex approach and the results using an unsteady BEM are therefore expected to be similar to the LLFVW output.

We thank the Reviewer for pointing this out. We were indeed aware of this but had not pointed it out in the paper. This consideration is now included in line 141 and 142 of the revised manuscript.

The axial induction velocities from the LLFVW shown in Figure 7 are very small and in the order of 0.03 m/s and compared to the inflow velocity of 4 m/s correspond to an axial induction factor of around a=0.03/4=0.008. If this is true then there is practically no induction for this case and what is the corresponding CT ?

The Reviewer is right. Due to a bug in the plotting script, a scale factor was erroneously being applied to induced velocities. Both figures 7 and 8 have been corrected and replaced. Thank you for catching this mistake.

On page 14 is reported a time constant for the dynamic wake LLFVW computations of around 3-4 seconds. In the Øye Dynamic Wake model the time constant is approximately the rotor diameter divided by the free wind speed and in the case of the UNAFLOW wind turbine should be around tau=2.4/4=0.6 seconds. That is the LLFVW model responds quite much slower to a dynamically changed force than will the Øye model. Since both the Øye dynamic wake model and LLFVW are based on a similar vortex ring model for the shed vorticity what is then the reason for this difference in time constant ? An Actuator Line simulation that through the N-S equations resolves the real physics and inertia of the wake response could be used to check these LLFVW results.

The Reviewer has a good point. The description the Reviewer is referring to was not precise. The time induced velocity takes to reach the new equilibrium value in Figs 7 and 8 is confused with rotor thrust. We have corrected the text (L360 and 379). In addition, to confirm the LLFVW results, we have run ALM simulations for the step tests in blade pitch and rotor speed. Although the magnitude of the step change is slightly different, and the overshoot of Fx slightly lower, the time the ALM simulations take to readjust to the new equilibrium thrust values is similar to LLFVW. The ALM results are included in Fig. 7 and 8 and in the text (L360-365)

The result shown in figure 12 is interesting. Here the simulations show that the axial induction factor for a pitching motion of amplitude between 1 and 2 degrees at a frequency of 0.1Hz and at a wind speed of 5 m/s can be as high as a=4 near the blade tip, meaning that the blade will experience a velocity from behind of about 3 times the wind speed. This is estimated to occur at wave heights of 10-13 meters. Is it a realistic scenario to have a wind speed of only 5 m/s at wave heights of more than 10 meters ? And how should a Glauert correction be when the axial induction becomes so large corresponding to a thrust coefficient way above 2 ? And is it the free wind speed or the apparent wind speed taking the structural velocity into account one should use when computing the thrust coefficient CT in a BEM based model ?

The Reviewer is again spot-on. Such high waves in low wind speed conditions are unlikely. We have highlighted this in the text, also in response to some internal members of our teams that have read the draft and raised similar concerns. We think this case represents a limit case, and is useful in this work to "stress test" the aerodynamic models and find their limits. However, despite being unlikely, we believe that this condition is not unrealistic. In a recent study, some of the authors have analyzed environmental conditions from a European site (Papi, 2022), which is known to have severe met-ocean conditions. We have found the 50-year extreme significant wave height at the site through statistical extrapolation techniques to be in the order of 8.5 meters at 5 m/s. The highest 10% of waves is generally 30% higher than the significant wave height and maximum wave height can be up to double the significant wave height. We have changed the text to reflect this in lines 495-503.
The BEM model we used in this study includes structural velocity in the momentum balance (to compute the thrust coefficient) and Buhl's (Buhl, 2005) implementation of the Glauert correction. This method has shown excellent agreement to the higher order theories in terms of its capability to predict global rotor forces (Fig. 11), proving that this approach, although theoretically incomplete, is still a

viable industrial tool. On the other hand, if we look at blade loads, especially for blade #1 that experiences the largest degree of relative inflow variation, BEM falls somewhat short of the higher order theories. This most likely indicates that there is some room to improve these models. It is possible that the effect of the blades moving through a varying induced velocity field, that the Reviewer brings up in the following point, may be relevant here. This study is already quite long, and we do not think it is the appropriate place to attempt to develop and test a new BEM formulation, but we have added a discussion in this regard based on the points raised by the reviewer in lines 543-548.

It is well known that BEM becomes inaccurate for large blade deflections for a bottom fixed wind turbine. This is because the blade elements are moved away from the rotor plane where the induction is computed combined with a strong streamwise gradient of the induced velocity near the rotor. This effect can become very severe in the case of a floating wind turbine where the position of the rotor plane is continuously moving along the wind direction and how to possibly treat this in a dynamic BEM code should also be discussed in a paper like this. The question is whether the induced velocity field follows the oscillating FOWT or the rotor moves in a velocity gradient fixed in space. This effect is very important and depends on the time constants of the rotor oscillation compared to the inertia (time constants) of the flow as also discussed in the paper by J.B. de Vaal et al.

We wish to thank the Reviewer for pointing this out. In the present AeroDyn, the momentum balance is performed separately on each blade element, in the blade-element reference system. Therefore, the streamtube effectively follows the blade position. As such, the blades do not move in a velocity gradient field that is fixed in space. The LLFVW and in the ALM models however are able to capture this effect in as much as it occurs. It is important to keep in mind (and the Reviewer is certainly aware) that, as stated in section 2.1, there is effectively no such thing as "rotor plane" or "streamtube" in AeroDyn, which is only loosely based on BEM as a theory and, in our opinion, fully qualifies as an Engineering Model. What we find interesting is that this implementation, despite being applied in conditions in which it should be invalid, works well in practice. This being said, we do believe the Reviewer has a point: if the oscillation frequency is high enough respect to the timescales and inertia of the flow, the wake will likely not have enough time to react and thus the rotor will effectively be moving in a velocity gradient fixed in space. We have included this consideration in the introduction in lines 69-72. In Fig. 4, we have shown results for varying oscillation frequencies. From a global rotor load perspective, we do not see the differences between BEM and higher order models increase significantly until we reach 8 Hz, at which point the returning wake effect becomes important. This is because, despite the timescale of the flow being around 0.6 s, the tested amplitudes are too small to appreciate this effect (0.33% of rotor diameter). In Fig. 12, however, this may play a role. We have included this in the discussion in lines 547-548.

**References**

Papi, F., Perignon, Y., and Bianchini, A.: Derivation of Met-Ocean Conditions for the Simulation of Floating Wind Turbines: a European case study, J. Phys.: Conf. Ser., 2385, 012117, https://doi.org/10.1088/1742-6596/2385/1/012117, 2022.

Buhl, M. L., Jr.: New Empirical Relationship between Thrust Coefficient and Induction Factor for the Turbulent Windmill State, https://doi.org/10.2172/15016819, 2005.

Boorsma, K. and Caboni, M.: Numerical analysis and validation of unsteady aerodynamics for floating offshore wind turbines, TNO, Delft, Netherlands, 2020.

Mancini, S., Boorsma, K., Caboni, M., Hermans, K., and Savenije, F.: An engineering modification to the blade element momentum method for floating wind turbines, J. Phys.: Conf. Ser., 2265, 042017, https://doi.org/10.1088/1742-6596/2265/4/042017, 2022.

Cioni, S., Papi, F., Pagamonci, L., Bianchini, A., Ramos-García, N., Pirrung, G., Corniglion, R., Lovera, A., Galván, J., Boisard, R., Fontanella, A., Schito, P., Zasso, A., Belloli, M., Sanvito, A., Persico, G., Zhang, L., Li, Y., Zhou, Y., Mancini, S., Boorsma, K., Amaral, R., Viré, A., Schulz, C. W., Netzband, S., Soto-Valle, R., Marten, D., Martín-San-Román, R., Trubat, P., Molins, C., Bergua, R., Branlard, E., Jonkman, J., and Robertson, A.: On the characteristics of the wake of a wind turbine undergoing large motions caused by a floating structure: an insight based on experiments and multi-fidelity simulations from the OC6 project Phase III, Wind Energ. Sci., 8, 1659–1691, https://doi.org/10.5194/wes-8-1659-2023, 2023.

---

## Author Response (AR2)

**Going Beyond BEM with BEM: an Insight into Dynamic Inflow Effects on Floating Wind Turbines**

F. Papi, J. Jonkman, A. Robertson, A. Bianchini

Dear Reviewer, dear Editor,

Thank you for your time managing and reviewing our work and for your qualified feedback. Based on the Reviewer' suggestions, we have done our best to improve the paper.

We have provided detailed answers to your comments below, in blue colored text for your convenience.

Best regards,

F. Papi, J. Jonkman,
A. Robertson, A. Bianchini

ooo ooo ooo

**Reviewer #1 – M. Hansen**

I am still not quite convinced that including the structural velocities from e.g. surge in the momentum equations is correct, unless the inertia term from accelerating the mass enclosed in the control volume is somehow also considered. But as written in the paper this is debatable.

The Reviewer is absolutely right. The approach of treating structural velocities as if they were variations in relative inflow is formally incorrect. When approaching this problem, we found it surprising that despite this, BEM shows remarkably good performance in many cases if compared to experiments and higher-order models. This is indeed the spirit with which this work was approached and the motivation behind the title "Going beyond BEM with BEM". That being said, we acknowledge that some paragraphs could have been phrased better, and thus the following was done to improve the paper:

1. Introduction: a paragraph was removed, as it partially overlapped with section 2.1 and could add confusion. When describing the paper objectives, the following was added: "In particular, the predictive capabilities of the—formally incorrect—approach of considering structural velocities as apparent inflow in the BEM balance and using a dynamic wake model to account for the changing conditions on the rotor in such a scenario are evaluated."

2. Section 3.2: the end of this section was significantly restructured, to better highlight the limitations of the approach of considering structural velocity in the momentum equation that emerged during the study.

3. Section 3.3: the end of this section was streamlined. Some of the considerations were moved to the conclusions, as they refer not only to the results presented in section 3.2 but to the information presented throughout section 3.

4. Section 4: conclusions were amended in many places, with the main rationale being to better highlight the performance of BEM in relation to how it is formulated (i.e., with structural velocities in the momentum balance). In particular, it is explained more clearly that this approach is found to work well in the tests at rated wind speed. Less so in LC 3.26, because such a model cannot distinguish between VRS and flow reversal on the rotor.

In my opinion a dynamic wake model as e.g. Oye model is modelling the inertia of the flow when changing the loads for a fixed rotor. An open question is, if it is good enough to apply such a model for a dynamically surging rotor and only consider the structural velocity in the blade element part and keeping the control volume fixed.

Thank you again for the interesting comment. Indeed, this remains an open question. In this paper, the problem is approached from a somewhat similar, but also quite different perspective: i.e., is it good enough to treat structural velocities as apparent wind variations, and how does a dynamic wake model handle such a scenario? We believe that the perspective of this paper is clearer thanks to the changes to the manuscript detailed within the previous point. We have highlighted the point of view the Reviewer raised in the conclusions: "Because structural velocities are included in the momentum balance as apparent wind variations, BEM and DBEM cannot distinguish between flow reversal and VRS, and therefore, despite the good prediction of aggregated rotor aerodynamic forces, differences with respect to higher order theories in the prediction of spanwise quantities are apparent. This highlights an area of possible future improvement in BEM-based engineering tools. For instance, the ability of a dynamic wake induction model to model the inertia in the flow of a floating rotor if structural velocity is not considered in the momentum balance, and possible improvements to these models in such a scenario, remains an open question."

In the same line I don't consider Eq. 1 on page 5 as being part of the momentum equation, but purely blade element theory as this is only an expression for the apparent inflow at the rotor plane. The momentum equation would be an equation to solve for the actual axial induction, a, in this expression.

This is true, Eq. 1 is not part of the momentum balance in a strict sense. However, it does model the contribution of the wake on the relative inflow at the rotor. The Reviewer is anyhow right in the fact that, as presented, it may introduce confusion. Therefore, we now refer to this equation as an expression for the "axial velocity at the rotor plane".

It could perhaps be mentioned that the conditions in Table 1 on page 9 correspond to the optimum TSR=7.4 for the DTU 10 MW reference turbine.

This is again a good observation. We have edited the caption of Table 1 accordingly.

On page 11 line 280 is stated that the ALM and the LLFVW only use static airfoil data. Why not include a dynamic stall model as this should give better results. Is there a reason for this ?

In the tests performed in this study, and during the UNAFLOW campaign at large, apart from the inner part of the blade, which is responsible only for a fraction of the rotor thrust, the angle of attack is below the stall angle. Therefore, it is important to capture unsteady attached-flow effects, rather than dynamic stall. Both of these effects are included in the dynamic stall models in OpenFAST, which are based on the Beddoes-Leishman dynamic stall model. In particular, attached-flow unsteady effects are modelled according to Theodorsen's theory, which contains a circulatory part, and an apparent mass part. The reduced frequencies at play in this study are, however, low, and therefore, the contribution of apparent mass effects is negligible. Circulatory effects are already embedded into ALM and LLFVW theory in

an indirect manner, as both models generate unsteady vortices in the wake of the blade. To avoid accounting for these effects twice, dynamic polars were not considered in the ALM and LLFVW models. We have improved the explanation on page 4: "In this work, inflow angles are mostly kept below stall, and thus attached-flow unsteady aerodynamic effects have the largest impact on results. These are mainly caused by two effects: added mass and shed vorticity, with the latter being by far the most relevant at the reduced frequencies analyzed in this work. The widespread consensus is that these effects are intrinsically included in higher-order aerodynamic theories such as LLFVW and ALM and do not require dynamic polars to model this effect. Therefore, static polars are used for the aerodynamic models in this study."

Various step cases are investigated, but is a step change in rotational speed relevant for a modern wind turbine having an enormous angular inertia moment making quick changes in rotational speed very difficult.

The Reviewer is right. As rotors increase in size, their rotational inertia also increases. In fact, if the IEA 15-MW rotor is compared to the NREL 5-MW rotor, the rotational inertia has increased more than 8 times, while swept area has increased a comparatively small 3.6 times. In the study, however, we are interested in evaluating how changes in surge and in rotational speed combine. Therefore, the step-changes that are simulated are purely conceptual and should be interpreted strictly in this sense. This has been also clarified in the paper.